# Revealing Genetic Diversity and Population Structure of Endangered Altay White-Headed Cattle Population Using 100 k SNP Markers

**DOI:** 10.3390/ani12223214

**Published:** 2022-11-20

**Authors:** Bo Liu, Weikun Tao, Donghe Feng, Yue Wang, Nazigul Heizatuola, Tenes Ahemetbai, Weiwei Wu

**Affiliations:** 1Key Laboratory of Genetics Breeding and Reproduction of Xinjiang Wool Sheep & Cashmere Goat, Institute of Animal Science, Xinjiang Academy of Animal Sciences, Urumqi 830000, China; 2Animal Husbandry Workstation in Altay Region, Altay 836000, China

**Keywords:** Altay white-headed cattle, genetic diversity, population structure, SNP chip, species in imminent danger

## Abstract

**Simple Summary:**

Under a low-input production system, local cattle have extraordinary adaptability to a variety of environments. However, due to various reasons, Native Altay white-headed cattle selection and breeding have not received enough attention. In addition, a large number of excellent foreign cattle varieties have been introduced, and unplanned hybridization has been carried out. The number of pure Altay white-headed cattle has been decreasing, with the breed even facing the danger of extinction. The genetic diversity index is of great significance for evaluating population structure and formulating conservation strategies. The genetic variation and population structure of Altay genetic resources were studied using 100 k SNP genotyping data from Altay white-headed cattle. Our results show that the genetic structure of the Altay white-headed cattle population is different, but the level of genetic differentiation is low, and the genetic diversity is low. In conclusion, our findings provide genetic information about the Altay white-headed cattle population, which can be used for future conservation and breeding research.

**Abstract:**

Understanding the genetic basis of native cattle populations that have adapted to the local environment is of great significance for formulating appropriate strategies and programs for genetic improvement and protection. Therefore, it is necessary to understand the genetic diversity and population structure of Altay white-headed cattle so as to meet the current production needs under various environments, carry out continuous genetic improvement, and promote rapid adaptation to changing environments and breeding objectives. A total of 46 individual samples of endangered Xinjiang Altay white-headed cattle were collected in this study, including nine bulls and 37 cows. To collect genotype data, 100 k SNP markers were used, and then studies of genetic diversity, genetic structure, inbreeding degree, and family analysis were carried out. A total of 101,220 SNP loci were detected, and the genotype detection rate for individuals was ≥90%. There were 85,993 SNP loci that passed quality control, of which 93.5% were polymorphic. The average effective allele number was 0.036, the Polymorphism Information Content was 0.304 and the minimum allele frequency was 0.309, the average observed heterozygosity was 0.413, and the average expected heterozygosity was 0.403. The average genetic distance of Idengtical By State (IBS) was 0.3090, there were 461 ROH (genome-length homozygous fragments), 76.1% of which were between 1 and 5 MB in length, and the average inbreeding coefficient was 0.016. The 46 Altay white-headed cattle were divided into their families, and the individual numbers of each family were obviously different. To sum up, the Altay white-headed cattle conservation population had low heterozygosity, a high inbreeding degree, few families, and large differences in the number of individuals in each family, which can easily cause a loss of genetic diversity. In the follow-up seed conservation process, seed selection and matching should be carried out according to the divided families to ensure the long-term protection of Altay white-headed cattle genetic resources.

## 1. Introduction

China has a vast territory with complex and diverse terrain, giving birth to many cattle breeds with local characteristics. For example, Xinjiang Altay white-headed cattle are favored by the majority of farmers and herdsmen in Xinjiang due to their excellent characteristics such as early maturity, strong reproductive ability, fast growth and development, high cold resistance, high temperature resistance, rough feeding resistance, good adaptability, and strong disease resistance. They also have good production capacity, tender meat, a high milk fat rate, a beautiful shape, and a gentle character, with ancient and primitive strains [1].

Worldwide, the cattle subfamily mainly includes six genera, 13 cattle species, and nearly 800 varieties. Due to the different times of origin and differentiation, the living environment and geographical distribution of different cattle breeds are also quite different. Altay white-headed cattle are an important breed of livestock in Altay. They make potential contributions to rural livelihoods, organic agriculture, and sociocultural and historical events. The number of cattle in the region has reached 733,000, with 525,100 in stock. The main cattle varieties are Xinjiang brown cattle, Holstein, Simmental, white-headed cattle, and Kazakh cattle [2]. Among them, there are 3900 Altay white-headed cattle in stock. They are mainly distributed in Buerjin County, with 3100 heads, and Habahe County, with 800 heads [3]. Altay white-headed cattle are a meat and milk dual-purpose breed, being the subject of a local unique variety conservation project in our region. The milk fat rate of Altay white-headed cattle in the region is 4.5%, the reproduction rate is 80%, and the improved variety rate is 25%. The average annual yield of fresh milk per head is 1 ton [4]. As Hemukanas Mongolian Nationality Township is far away from other pastures and has a special geographical location that is conducive to the conservation of white-headed cattle, the Altay white-headed cattle conservation area is designated in Hemukanas Mongolian Nationality Township, and male and female cattle of other varieties are strictly prohibited from entering the conservation area to prevent group mixing [5].

Genetic diversity is an important component of biodiversity. Generalized genetic diversity refers to the sum of various pieces of genetic information carried by organisms on the earth. In the narrow sense, genetic diversity is mainly the variation in intraspecific genes, including the genetic variation between different populations and within the same population [6]. Genetic diversity illustrates a key aspect of differentiation between individuals of a population that exists at the phenotypic or DNA level. Genetic diversity indicators provide important information for the conservation and improvement strategies of domestic animal populations and their adaptation to specific environments. With the emergence of SNP chip data, the field and depth of bovine genome research have undergone new developments. Genome-wide SNP chips have been used to study the genetic diversity, population structure, inbreeding level, exozygosity analysis, effective population size, linkage disequilibrium, migration events, genome-wide association research, and selection feature detection of different cattle populations around the world. Additionally, using SNP chip data to estimate the value of genome breeding and its prediction accuracy is another frontier approach in the process of genetic evaluation. On the other hand, compared with traditional family data, more accurate diversity parameters can be obtained by using genomic information. However, the population of Altay white-headed cattle is still lacking research, especially at the molecular level. In recent years, SNP marker analysis has become a standard method for diversity analysis and genome-wide research. SNP markers represent a more interesting genotyping method because they are abundant in the genome, genetically stable, and easy to use in high-throughput automatic analysis [7]. Some studies have demonstrated the effectiveness of SNPs in population diversity and structure analysis [8]. Although a large number of SNPs were found in the bovine genome sequencing project, they have not been verified in the Altay white-headed cattle population before. Variety characteristics require basic genetic variation knowledge, which can effectively measure genetic variation within and among populations. Therefore, the purpose of this study is to explore the degree of genetic diversity, population structure, and differentiation of endangered Altay white-headed cattle. In fact, genetic diversity is of great significance for genetic improvement and rapid adaptation to changing environments and breeding goals [9].

## 2. Materials and Methods

### 2.1. Main Test Reagents

The CWE9600 Maghead Blood DNA Kit from China Jiangsu Kangwei Century Biotechnology Co., Ltd., Taizhou, China, red blood cell lysate (RT122), 6× Loading Buffer, molecular weight marker, fast agar gel DNA recovery kit, Hot Start Taq polymerase, agarose, and anhydrous ethanol were purchased from China Beijing Tiangen Biochemical Technology Co., Ltd., Beijing, China.

### 2.2. Test Animals

Blood samples from the jugular vein of 46 Altay white-headed male and female cattle, including 37 cows and nine bulls, were collected in the Hemukanas Mongolian Nationality Township, Altay City, Xinjiang Province, and stored in a refrigerator at −80 ℃.

### 2.3. Test Method

#### 2.3.1. DNA Extraction

The magnetic bead method was used to extract DNA. Based on the principle of specific adsorption of DNA by silicon-based magnetic beads, the cwe9600 magbead blood DNA kit was used to release the blood genome through enzymatic hydrolysis, bind DNA molecules in a specific high-efficiency lysis buffer system, and change the environmental ionic strength so as to realize the isolation and purification of genomic DNA.

#### 2.3.2. DNA Quality Detection

After extraction, a 2% agarose gel was used for electrophoresis to detect the quality, a gel imager was used to observe the size and brightness of genomic DNA fragments, a Qubit was used to accurately quantify the DNA concentration, a Nanodrop was used to detect the purity of DNA samples (OD260/OD280 = 1.7–2.1), and photos were taken for recording.

#### 2.3.3. SNP Typing of Genomic DNA

To extract qualified gDNA (concentration ≥ 50 NG/UL), first the whole genome of all samples was amplified (the PCR amplification program was set to 35 cycles, with denaturation at 94 °C for 30 s, annealing at 55 °C for 30 s, and extension at 72 °C for 1 min). Afterwards, the gDNA was incubated at 37 °C for 20–24 h, and then it was fragmented, precipitated, and resuspended in hybridization buffer. The resuspended DNA fragments were added to the chip, hybridized, and incubated at 48 °C for 16–24 h. After hybridization, the non-specific binding DNA was removed by washing, and the specific binding sites were left for single-base extension. After staining, the data were scanned and read with the Illumina iSCAN reader for data analysis.

#### 2.3.4. Genetic Diversity Analysis

Genetic diversity analysis includes the effective population size (Ne), which refers to the ideal population content with the same gene frequency variance or the same inbreeding coefficient increment (heterozygosity decay rate) as the actual population [10]. It is usually estimated based on the linkage disequilibrium (LD) level of the population [11]. To calculate the proportion of polymorphic markers (PN), we first calculated the minimum allele frequency (MAF) of each locus using Plink software [12], and then we calculated and analyzed the proportion of polymorphic markers. The expected heterozygosity (He) of a population refers to the probability that any individual in the population is heterozygous at any point, and the observed heterozygosity (Ho) refers to the proportion of the number of individuals who are heterozygous at a certain point in the population to the total number of individuals. The Polymorphism Information Content (PIC) is an index for measuring the degree of gene variation and reflecting the amount of genetic information. PIC can be calculated according to the formula of Bostein et al. [13]. The number of effective alleles and the MAF refer to the frequency of uncommon alleles in a given population. The Plink (v1.90) software was used to analyze the proportion of polymorphic markers, He and Ho, and the SNeP (v1.1) software was used to analyze the effective content of the population.

#### 2.3.5. Inbreeding Coefficient Analysis Based on Long Homozygous Fragments

Plink1.9b software was used to analyze runs of homozygosity (ROHs). First, Plink (v1.90) was used to calculate the ROH length of each sample. The inbreeding coefficient based on ROHs was calculated by calculating the proportion of the total length of ROH fragments in an individual to the total length of the autosomal genome [14]. Therefore, the longer the total length or the higher the number of ROHs in an individual, the higher the inbreeding coefficient of that individual.

#### 2.3.6. Genetic Relationship Analysis of Population Genome

The genomic relationship G matrix was constructed using the genome-wide marker information, and the G matrix molecular kinship analysis was carried out using GCTA (V1.94) software. A heat map was drawn to show the genomic kinship among individuals [15]. The Plink (v1.90) software was used to calculate the genetic distance between individuals, construct an identity-by-state (IBS) matrix, and analyze the genetic distance of IBS.

#### 2.3.7. Cluster Analysis to Construct Altay White-Headed Cattle Family

Cluster analysis is a method of generating a relatively simple class structure from a group of complex data and classifying groups according to the degree of correlation or similarity between different individuals [16]. We used the adjacency method (neighbor joining, NJ) and clustered the samples based on the genetic distance matrix obtained from the genetic distance analysis. Through clustering analysis, closely related individuals were aggregated into a small taxon, and relatively distant individuals were aggregated into a larger taxon, until all the samples were aggregated. Finally, the whole classification system was turned into a genealogy chart, which shows the kinship between all samples and constructs the Altay white-headed cattle conservation population family.

## 3. Results

### 3.1. Genomic DNA Detection

The concentration of the 46 Altay white-headed cattle DNA samples was more than 100 ng/μ 50. The Od260 nm/od280 nm was 1.7~2.0. The genomic DNA samples were of good quality, without protein pollution and degradation Figure 1. Thus, the samples meet the analysis requirements of gene chips, meaning the samples could be used for subsequent research.

### 3.2. SNP Quality Control and Typing

#### 3.2.1. Sample Quality Control

In order to ensure the accuracy of the analysis, Plink (v1.90) software was first used to check the duplicate samples. The criterion for judging the duplicate samples was as follows: when the DST between samples was ≥0.99, the samples were judged as duplicate samples. DST is the probability that two individuals reflect homomorphism at the genomic level. Two pairs of duplicate samples were found in this analysis. We reserved the samples with a high detection rate for analysis.

#### 3.2.2. SNP Locus Quality Control

Plink (v1.90) software was used for the quality control of SNP loci, and only the loci with the best typing quality were reserved for subsequent analysis. Table 1 presents the specific quality control conditions and results, and Figure 2 presents the distribution of SNPs on each chromosome before and after quality control.

### 3.3. Genetic Diversity Analysis

The results of the genetic diversity analysis are shown in Table 2. A total of 1.704 effective alleles were detected in 46 Altay white-headed cattle, with an average effective allele number of 0.036 and a minimum allele frequency of 0.309. Among them, there were relatively many between 0.4 and 0.5, accounting for 33.09%. The distribution is shown in Figure 3a. The polymorphism marker ratio (PN) of SNP sites was 0.935, indicating that 93.5% of the SNP sites were polymorphic, and the Polymorphism Information Content of the SNP sites was 0.304, as shown in Figure 3b. The effective population content (Ne) of Altay white-headed cattle was 2.4, the average Ho was 0.413, and the average He was 0.403. The average Ho was slightly larger than the average He, but the two were very close, as shown in Figure 3c.

### 3.4. Inbreeding Coefficient Analysis Based on Long Homozygous Segments

A total of 461 ROH fragments were detected in the conservation population of 46 Altay white-headed cattle, with the largest number of ROHs being between 1 and 5 MB in length, accounting for 76.1%, as shown in Figure 4a. Among them, the shortest ROH was 1.01 mb in length, located on chromosome 18, and the longest ROH was 62.29 mb in length, located on chromosome 20. Among them, the number of ROHs on chromosome 4 was the largest, with 31, and the number of ROHs on chromosome 27 was the smallest, with 2, as shown in Figure 4b.

The number of individuals with a total length of ROHs between 0 and 50 MB was the largest, with 32 heads in total, as shown in Figure 4c, accounting for 69.6%. The average inbreeding coefficient calculated based on the ROHs of the Altay white-headed cattle conservation population was 0.016, the lowest was 0.015, and the highest was 0.1106, indicating that the Altay white-headed cattle conservation population is inbred, as shown in Figure 5.

### 3.5. Genetic Relationship Analysis of Population Genome

#### 3.5.1. Kinship Analysis Based on G Matrix

In this experiment, a total of 101,220 SNP loci were detected in 46 samples, and 85,993 SNP loci after quality control were used for principal component analysis. It can be seen from Figure 6 that Altay white-headed cattle have a moderate genetic relationship and relatively high diversity. However, in the first principal component, the individuals on the right are closely related, indicating that there may be inbreeding. Then, the G-matrix of the population (Figure 7) was constructed to further analyze the genetic relationship of the Altay white-headed cattle conservation population, and the results of principal component analysis were also verified. Most Altay white headed cattle showed a moderate to high degree of genetic relationship among individuals (the lighter squares in Figure 7), while some individuals had a closer genetic relationship (the darker squares in Figure 7), indicating that there was a certain inbreeding trend and that this sample should be avoided in the breeding plan.

#### 3.5.2. Genetic Distance Analysis of IBS

The genetic distance of IBS among the 46 Altay white-headed cattle was between 0.1927 and 0.3451, with an average of 0.3090. The results of the IBS distance matrix of the Altay white-headed cattle conservation population are shown in Figure 8. Most of the squares are light in color (showing a moderate degree of kinship). The IBS genetic distance between a small number of individuals is close, and there is high kinship.

### 3.6. Cluster Analysis to Construct Altay White-Headeded Cattle Family

#### Cluster Analysis

In view of the importance of bulls to the whole conservation population, we extracted samples of bulls and conducted cluster analysis separately to judge their kinship. The results are shown in Figure 9, and the cluster analysis results for all samples are shown in Figure 10.

### 3.7. Family Structure Analysis of Altay White-Headed Cattle Conservation Population

Combined with the results of the genomic kinship analysis and cluster analysis, the existing bull samples were divided into one family. The standard for clustering is that the genomic kinship coefficient between bulls is greater than or equal to 0.1. According to the kinship between the existing cows and bulls, different families were established. In addition, it was found that 30 cows were distant from the detected bulls. Therefore, they were classified into the “other” category (see Table 3 for the results).

## 4. Discussion

The native Altay white-headed cattle population has mostly non-selective genetic resources as a result of random reproduction, forming a unique gene pool due to its adaptation to the local environment [17]. Due to the lack of strong directional selection for production traits and the complex evolutionary background, a large number of varieties have been produced. Their phenotypes are well adapted to various environments and feeding systems, as well as different production purposes [18]. More diverse alleles are expected in these populations. In this study, we analyzed the whole-genome SNP data to improve our knowledge of the genetic structure and diversity of Altay white-headed cattle.

Most efforts have been focused on superior commercial varieties, while local varieties have not been sufficiently studied, although they can represent excellent genetic resources for the local economy [19]. At present, there is no report on the genetic diversity of Altay white-headed cattle.

### 4.1. Genetic Diversity Analysis

Altay white-headed cattle are one of the important local cattle breeds in the Altay region of Xinjiang. The number of existing pure-bred Altay white-headed cattle in this region is about 3000, making them an endangered species. Because their pedigree record has been imperfect, this has resulted in a pedigree error or an absence of pedigree results. In this study, based on SNP chip data, the existing populations of Altay white-headed cattle were evaluated as a whole, and the families constructed by the chip data were compared with the recorded pedigrees, helping to accurately understand the current genetic diversity and conservation status of Altay white-headed cattle. In the past 50 years, cattle breed diversity has experienced a serious decline, mainly due to the worldwide adoption of a small number of high-yielding breeds and intensive selection [20]. Therefore, evaluating the genetic diversity of cattle is an important step in cattle breeding management [21]. Scientific evaluation of the conservation effect is an important part of conservation work. At present, most studies use microsatellite marker technology to analyze the genetic diversity of conservation populations and evaluate the conservation effect [22]. Due to the recent emergence of high-throughput genotyping technology and microsatellite markers, it has become possible to carry out detailed genome-wide analysis of the genetic structure of cattle and the relationships between populations. As part of the genome selection revolution in animal husbandry and the introduction of genome methods in endangered population protection programs, these technologies have opened a new perspective for livestock genetics [23]. Improving our understanding of the diversity within breeds and the relationships and structures between breeds is the basis for improving selection design, breeds, and the efficient use of breeds, as well as understanding environmental adaptability and implementing conservation plans [24].

Abdelmanova et al. [25] selected nine microsatellite loci to describe the genetic diversity and relationship between historical and modern populations of Russian indigenous cattle breeds. No significant difference in genetic diversity between historical and modern representative breeds was observed, indicating that the conservation method used in this study sufficiently preserved the genetic diversity of Russian cattle. Compared with microsatellite markers, genome-wide SNP markers can more objectively reflect the genetic differences between individuals and are increasingly used to analyze the genetic diversity of populations [26]. Bhuiyan et al. [27] used 50 k SNP markers to reveal the heterozygosity estimation of indigenous cattle populations in Bangladesh, as well as NJ phylogenetic tree analysis. The overlapping clustering showed that the genetic differentiation between indigenous cattle populations in Bangladesh was weak. The effective population size indicates that the ancestors of Shaxihua and North Bengal gray cattle are limited. Zhang et al. [28] used a high-density SNP chip to reveal the population structure and mixing pattern level of Chinese indigenous cattle, revealing the origin and evolutionary history of these breeds. The analysis results showed that there was a history of central breed mixing in Chinese indigenous cattle, indicating an introgression event of Chinese indigenous cattle. Chinese indigenous cattle are divided into two genetic groups, corresponding to the northern indigenous cattle (Eurasian taurine line) and the southern indigenous cattle (Asian indicine line). SNP chip technology is widely used to analyze the genetic diversity of populations with high accuracy and low cost. Studies have shown that SNP markers are more accurate than microsatellite markers in estimating kinship [29].

The estimation of Ne in this study relies on Sved’s method of calculating the linkage degree between SNPs and the Morgan distance. Since the r2 of all SNP marker pairs at this physical length is then calculated, the accuracy of Ne estimation may be affected because SNP markers are not uniformly distributed [30]. In addition, the sample population size will also significantly affect the estimate of Ne [31]. Only 46 Altay white-headed cattle were used in this study, which is a smaller number than that used in other studies of the same breed, resulting in lower results [32]. However, the actual situation of species or varieties in the process of evolution and breeding is often much more complex, so the estimated value in this study is only a reference value that is as accurate as possible, and a more accurate estimation depends on the optimization of mathematical models and a better test population. This study found that the expected heterozygosity (He 0.403) of the Altay white-headed cattle population was less than the observed heterozygosity (Ho 0.413), indicating that the tested Altay white-headed cattle conservation population may have been differentiated in history or that there may have been an introduction of a small part of foreign blood, which needs to be further purified. Chen Zhihua et al. [33] used 12 microsatellite markers to detect the genetic diversity of six cattle breeds. The PIC value of the population ranged from 0.3444 to 0.5521. The average PIC value of Daegu cattle was the lowest (0.3444), indicating moderate polymorphism, and other cattle breeds showed high polymorphism. This study found that 93.5% of the SNPs were polymorphic, with an average PIC of 0.304, indicating moderate polymorphism. The PIC value in this study was slightly lower, indicating that the genetic diversity of Altay white-headed cattle was lost during the conservation process, indicating that conservation measures need to be strengthened. According to the analysis of the gene chip results, we can create a reasonable breeding plan to reduce inbreeding and maintain the genetic diversity of white-headed cattle without reducing the number of existing families in Altay.

### 4.2. Inbreeding Degree Analysis

ROHs are usually used to estimate the inbreeding coefficient of a genome. The length of ROHs and the proportion of the genome covered by ROHs can sufficiently reflect the age and origin of inbred lines, thus reflecting the level of inbreeding. The length of ROHs is in direct proportion to the genetic relationship between individuals. The longer the ROH segment, the greater the possibility of inbreeding, and vice versa [34]. The larger the number of ROH fragments, the more likely it is that there is inbreeding in this population [35]. A total of 461 ROH fragments were detected in the 46 Altay white-headed cattle populations tested in this study. The ROH length was below 5 Mb, accounting for 76.1%; the shortest ROH was 1.01 Mb; and a few individuals had an ROH length of more than 400 Mb. Altay white-headed cattle have the largest number of ROH individuals, with 0~50 ROHs, and the average inbred line value was 0.016. Yang Zhancheng et al. [36] used high-density SNP markers to analyze the genomic inbreeding of Chinese Holstein cattle. The results showed that 44,676 ROH fragments were detected, whose lengths were mainly distributed between 1 and 10 Mb. The ROHs of different lengths were scattered in the individual genome. Short ROHs and long ROHs were more common, and ROHs were not evenly distributed on the chromosomes. The region with the highest ROH frequency was the middle of chromosome 10. The correlation between the inbreeding coefficients of the two genomes was very high (more than 91%), but the correlation between genomic inbreeding and pedigree inbreeding was low (less than 50%). Kim et al [37] using a 50 k chip found that there were also some differences in the distribution of ROH in different populations, which reflects the genomic characteristics of the population. Honghao et al. [38] used 50 k SNP markers for genotype detection and found that Jinnan cattle have a distant relationship with Holstein cattle, Heshun beef cattle, Simmental cattle, and Limousin cattle in genetic structure, but are close to Yanbian cattle. Jinnan cattle are a Chinese local breed population. The carcass weight breeding value of Jinnan cattle reserve bulls was evaluated to assist in the selection of bulls with high breeding value, which improved the accuracy of selection, greatly saved time and cost, and could accelerate the genetic progress of Jinnan cattle. Shi Rui et al. [39] used SNP chip information to evaluate the homozygous degree of the Xinjiang inbred cattle genome. The results showed that the genetic background of Xinjiang inbred cattle was basically the same as that of Kazakh cattle. The homozygous degree of the inbred cattle genome was significantly higher than that of other populations; the higher the homozygous rate of the gene, the smaller the size of inbred cattle. To a certain extent, this shows the effect of inbreeding depression on body shape, which proves that inbreeding can cause variety depression and that the application of molecular-assisted selection can avoid inbreeding [40]. Zhang et al. [41] used 50 K chip and sequencing data to study Holstein cows, and evaluated the homozygosity and inbreeding level between individuals so as to guide artificial insemination and selection.

Due to the limitation of the population size and the relatively closed operation mode of Altay white-headed cattle in China, with the prolongation of the conservation time and the aggravation of generation overlap, an increase in the inbreeding coefficient and a change in genetic structure in the population are inevitable. In the later stage, the breeding plan can be reasonably arranged using SNP data, and the breeding method can be changed if necessary. Artificial insemination and other methods are used to ensure the stability of the population structure.

### 4.3. Analysis of Genetic Relationship and Genetic Structure

In this study, a total of 46 Altay white-headed cattle were detected using a 100 k cattle genome chip. Through NJTree and cluster analysis, it was found that Altay white-headed cattle could be grouped into one group in genetic clustering, which was different from domestic and foreign cattle in genetic structure. Its germplasm resources were unique, providing theoretical support and the basis for Altay white-headed cattle conservation. Wang et al. [42] analyzed the population genetic relationships of Jinnan cattle using the microsatellite marker method and found that Jinnan cattle are relatively closed in the breeding process and that there is not much crossing between the blood relationships with other cattle species, which have typical geographical limitations, retaining the independence of population genetic characteristics. This feature has certain similarities with Altay white-headed cattle. Ocampo et al. [43] used molecular markers to study the genetic diversity of Colombian Creole cattle. They genotyped 20 microsatellite loci of six local breeds and one introduced breed and found that Colombian Creole cattle breeds maintained a high level of genetic differentiation in the same population (93%), while the level for the rest of the breeds could be explained by differences between them (7%). The differentiation patterns and genetic relationships among Colombian Creole cattle breeds are highly consistent with the evolutionary history of each breed. They can be applied to auxiliary seed conservation, greatly improving efficiency and saving costs. Utsunomiya et al. [44] studied the genetic structure of zebra, buffalo, and extinct European bison herds in West Asia and South Asia, guided the conservation of cattle herds, and realized the continuation of valuable genetic resources. Acosta et al. [45] analyzed the genetic structure of five cattle populations in Cuba using microsatellite markers and found that the five cattle populations could be divided into two large groups, which made the goal of conservation clearer and more effective and played an important scientific guiding role in the whole-genome preservation of livestock species. Michailidou et al. [46] used a gene chip to study the genetic differences between two local populations of Greek sheep, thus providing a theoretical basis for seed conservation. Determining the population structure and genetic relationships has proved useful in conservation planning and developing appropriate management measures [47]. These studies have applied the cluster analysis method of modern genetic markers, providing theoretical support for the conservation of livestock and poultry resources.

In summary, a large number of genetic variations have been retained in Xinjiang Altay white-headed cattle. An analysis of the genetic distance, phylogenetic tree, principal components, and population structure clearly distinguished cattle populations with different historical origins, representing the genetic differences between the Altay white-headed cattle population and other Xinjiang varieties. The high genetic diversity within the population and the unique adaptation of the current population to a wider range of environmental factors (disease, heat stress, drought, and feed shortage) may be the result of special mixing between different cattle breeds. Thus far, the Altay white-headed cattle population has unique genetic resources and untapped opportunities, and measures need to be taken to protect and utilize this population sustainably.

### 4.4. Develop Ment of Appropriate Breeding Programmes

The average inbreeding coefficient of this group was 0.016. In order to reduce the inbreeding increment in the population, it is recommended not to breed bulls and cows of the same family. The genetic relationship coefficient between the 30 cows in the “other” category in the family group construction results and all the bulls was less than 0.1, meaning that these cows can be mated with any bull. After the genetic coefficient is confirmed, breeding can be carried out. An effective way to reduce the inbreeding coefficient is to expand the population content and carry out appropriate selection while strengthening management to avoid incomplete or incorrect pedigree records caused by inbreeding. Breeding bulls with large genetic distances can be selected to increase the effective population size. This study also found that the Altay white-headed cattle conservation group was only composed of three families, and the number of male and female cattle in the three families was quite different, with an unbalanced family structure. Therefore, it is necessary to introduce new lineages from other regions, expand the core group, and implement reasonable selection to avoid inbreeding and maintain the genetic diversity of the Altay white-headed cattle conservation population.

## 5. Conclusions

In this study, we found that the genetic diversity of the Altay white-headed cattle population was very low through the detection and analysis of a 100 k gene chip, especially according to the genetic coefficient, where there was only one family left for the bulls. The genetic cluster analysis of Altay white-headed cattle proved the uniqueness of Altay white-headed cattle in molecular genetic structure. This helps to understand the endangered situation of this population, which is conducive to our emergency protection of endangered species. Once the genetic distance between bulls and cows is known, first-generation bulls and cows selected for mating should have a relatively large genetic distance. In each generation, the genetic distance between individuals is determined to ensure that bulls and cows with the farthest genetic distance can produce offspring by mating, so as to ensure that the genetic diversity of the population is maintained at the maximum level. The selection accuracy of bulls can be improved through the combination of multiple breeding methods. This lays the foundation for the improvement of Altay white-headed cattle breeding.

## Figures and Tables

**Figure 1 animals-12-03214-f001:**
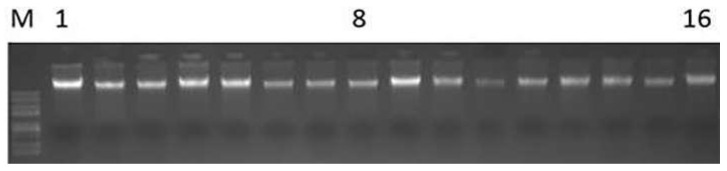
DNA gel electrophoresis.

**Figure 2 animals-12-03214-f002:**
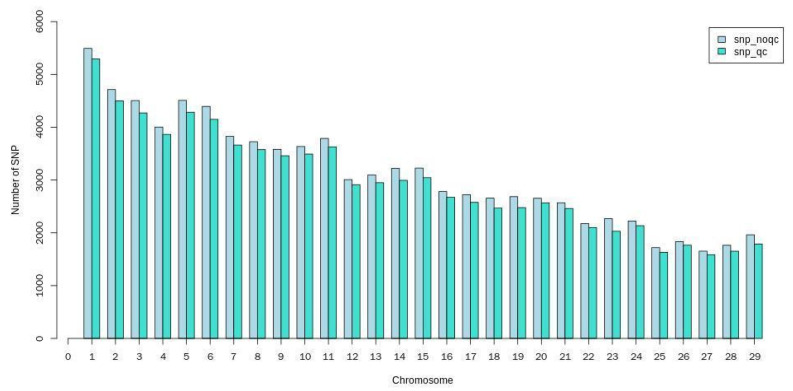
Distribution of SNPs on each chromosome before and after quality control. Note: the abscissa indicates the chromosome number, and the ordinate indicates the number of SNPs.

**Figure 3 animals-12-03214-f003:**
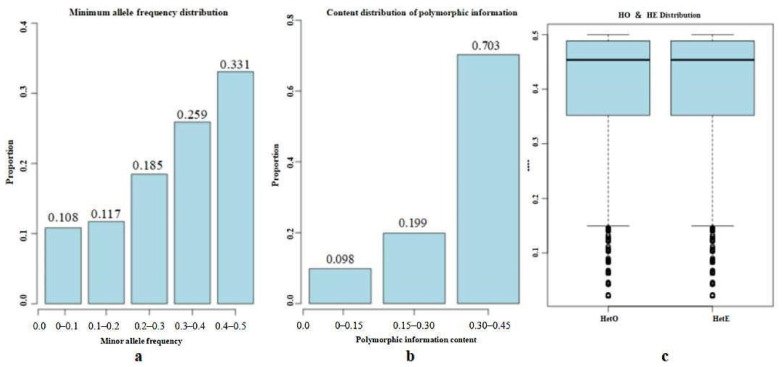
Genetic Diversity Analysis results. Note: (**a**). Minimum allele frequency distribution (the abscissa indicates the minimum allele frequency interval, and the ordinate indicates the SNP proportion). (**b**). Distribution of the Polymorphism Information Content (the abscissa represents the PIC interval value, and the ordinate represents the SNP proportion). (**c**). Heterozygosity analysis (Note: the abscissa indicates the classification of Ho and He, and the ordinate indicates the heterozygosity value).

**Figure 4 animals-12-03214-f004:**
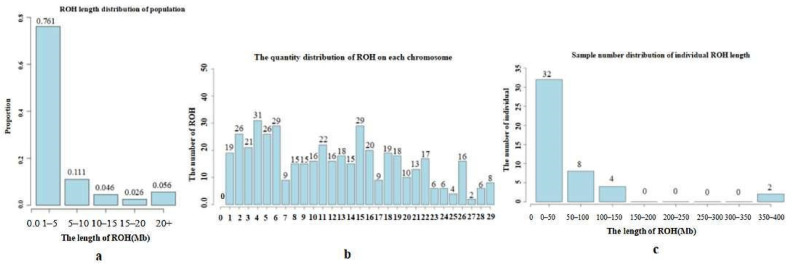
ROH analysis results. Note: (**a**). Distribution of Roh length by population (the abscissa represents the length interval of ROH, and the ordinate represents the population proportion). (**b**). Distribution of the ROH number on each chromosome (the abscissa represents chromosome number, and the ordinate represents ROH quantity). (**c**). Sample number distribution of individual ROH length (the abscissa represents the length interval of Roh, and the ordinate represents the number of individuals).

**Figure 5 animals-12-03214-f005:**
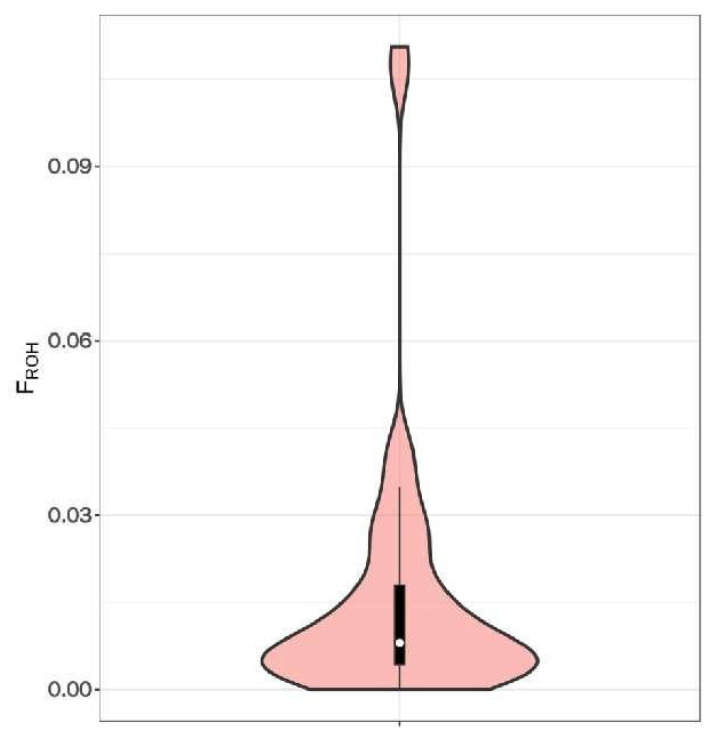
Distribution of the inbreeding coefficient froh based on Roh. Note: This violin chart is mainly used to show the distribution of data. The white dot in the center represents the median of the population FROH, and the upper and lower edges of the black box in the middle represent the upper and lower quartiles of the population FROH, respectively. The width of the violin chart indicates the probability density distribution of the population FROH. The wider part of the violin chart indicates that there are a larger number of samples at this level, and vice versa.

**Figure 6 animals-12-03214-f006:**
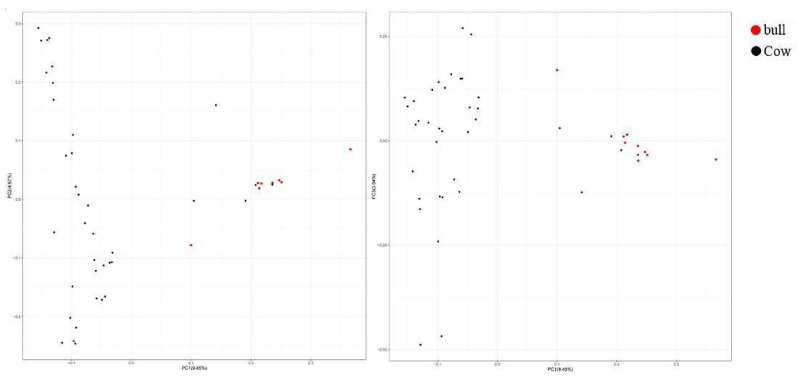
Principal component analysis results.

**Figure 7 animals-12-03214-f007:**
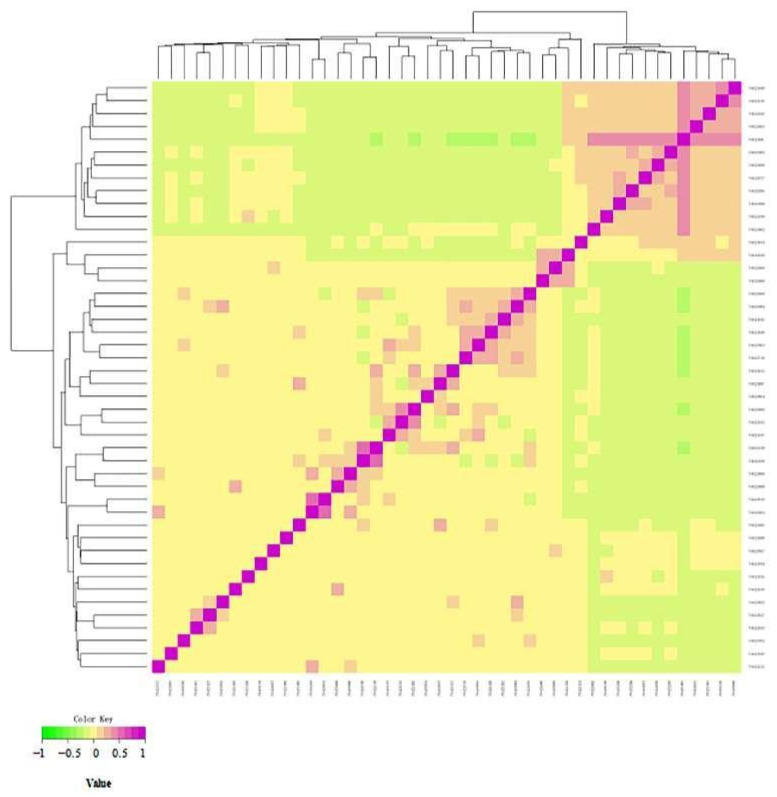
Visualization results of the genome kinship analysis. Note: This figure shows the genomic kinship coefficient between two individuals. The closer the color, the closer the kinship. The abscissa and ordinate indicate the individual IDs.

**Figure 8 animals-12-03214-f008:**
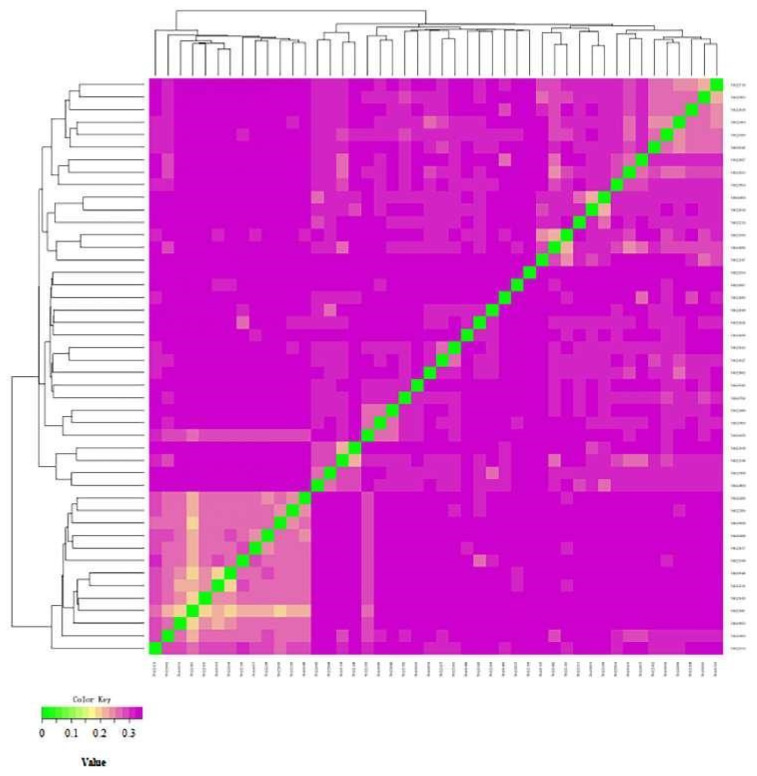
Visualization results of the genetic distance analysis. Note: This figure shows the genetic distance between two individuals. The closer the color is, the closer the genetic relationship is. The abscissa and ordinate represent the individual IDs.

**Figure 9 animals-12-03214-f009:**
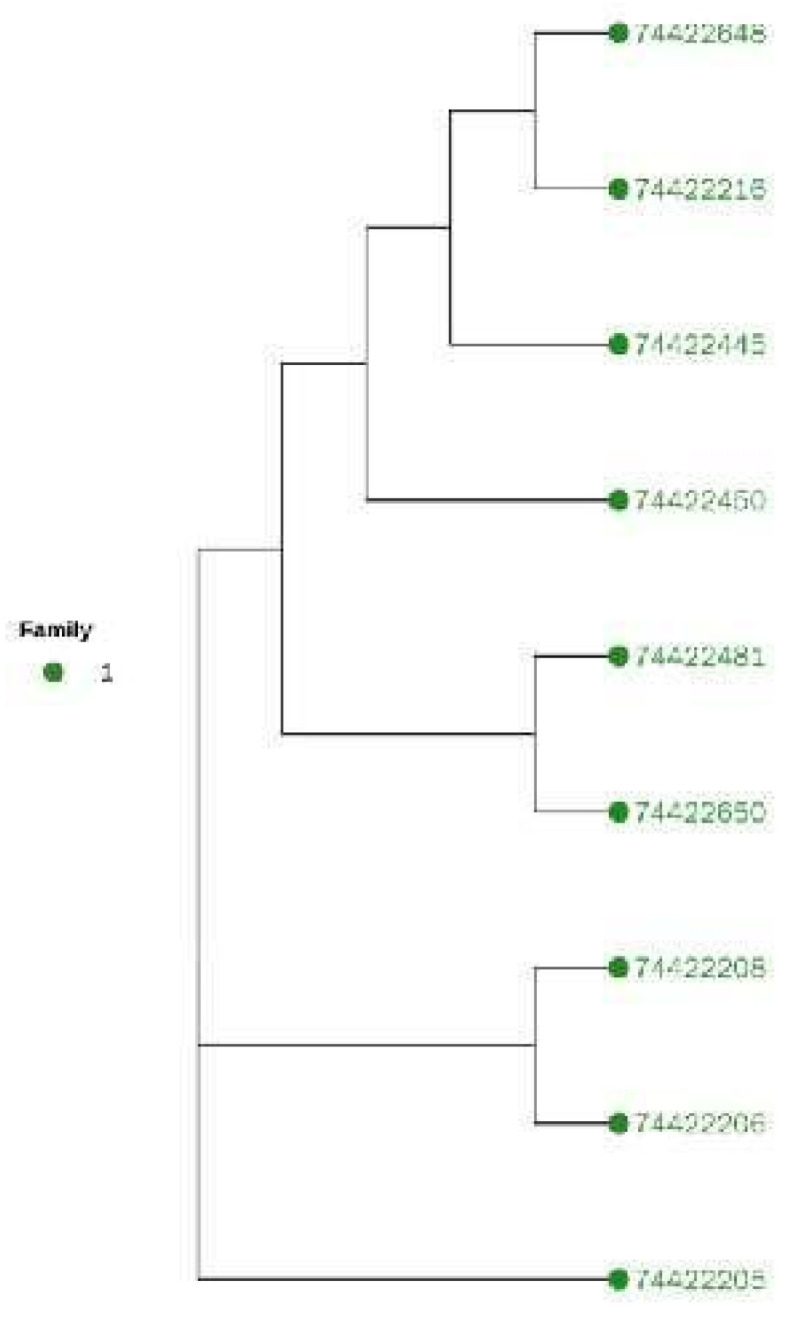
Cluster analysis results of bull samples. Note: one color represone family.

**Figure 10 animals-12-03214-f010:**
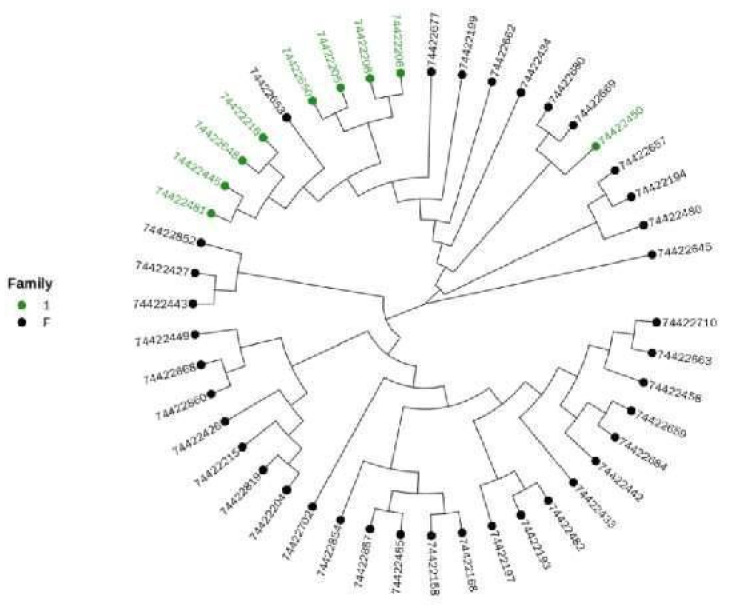
Clustering analysis results of all samples. Note: the color marked in the evolutionary tree is the bull sample, and one color represents a family.

**Table 1 animals-12-03214-t001:** Statistical Results of SNP Quality Control.

Quality Control Standard	Number of SNPs
Total number of SNPs	101,220
SNP with MAF < 0.01	3157
SNP not in Hardy–Weinberg equilibrium *p* < 10^−6^	66
SNP with callrate < 0.90	1010
SNPs on chromosome X	3785
SNPs on chromosome Y	330
SNPs on chromosome 0	6267
insertion/deletion	174
SNPs used after quality control	85,993

**Table 2 animals-12-03214-t002:** Analysis results of population genetic diversity.

Effective Population Content (Ne)	2.4
Proportion of Polymorphic Markers (PN)	0.935
Expected Heterozygosity (HE)	0.403
Heterozygosity Observed (HO)	0.413
Polymorphism Information Content (PIC)	0.304
Effective Numbers of Alleles	1.704
Minor Allele Frequency (MAF)	0.309

**Table 3 animals-12-03214-t003:** Results of population and family construction.

Family 1 Male Altay White-Headed Cattle	Family 1 Female Altay White-Headed Cattle	Miscellaneous	
74422650	74422199	74422645	74422702
74422481	74422677	74422657	74422668
74422205	74422653	74422480	74422854
74422648	74422662	74422194	74422819
74422206	74422434	74422426	74422204
74422216	74422669	74422449	74422857
74422208	74422680	74422443	74422442
74422445		74422485	74422482
74422450		74422852	74422458
		74422197	74422158
		74422168	74422684
		74422193	74422433
		74422427	74422659
		74422860	74422663
		74422215	74422710

## Data Availability

All data generated or analyzed during this study are included in this published article.

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
