# Peer review of "Revealing Genetic Diversity and Population Structure of Endangered Altay White-Headed Cattle Population Using 100 k SNP Markers"

_animals, 2022, doi:10.3390/ani12223214_

Round 1

Reviewer 1 Report

The manuscript described the genetic diversity and population structure of a native and endangered cattle namely Altay White Head Cattle.  The study is very important for conservation management activities. The manuscript needs correction. Points are indicated in the manuscript file (PDF).

Reviewer 2 Report

Dear authors, 

After analyzing submitted paper, I am sorry to see discrepancy between applied methods and obtained results, and especially conclusions. You have done good job by analyzing the data and choosing methods and techniques; however, results, discussion and especially conclusions, do not reflect this job. So, please improve according to instructions in the manuscript. Please, improve discussion, especially for effective population size and other parameters. More important, improve conclusion and provide the most important findings and implications of the paper. 

Reviewer 3 Report

Dear Authors,

unfortunately I can accept the manuscript in this form.

The aim of the study is valid and interesting, such as the type of genetic analyses performed in the survey. I comprehend that you have paid attention to an endangered breed, but I fear that for this type of study (a sort of Genome wide association study-GWAS) the number of analyzed animals is too low to be accepted (n=46). 

Furthermore, the quality of presentation is poor for English (i.e: a mix of verb forms, that is bothersome) but also for clarity of scientific significance. There are repetition of redundant concepts, mistakes in punctuation, lacking of instruments and company, lacking of reagents used for amplification PCR analysis, and other inaccuracies. Figures are too many, but moreover some are not very clear. 

The introduction is ok (my only suggestion is to add some deeper information about endangered breeds in your Country and about SNPs associated to different animals conditions, among them disease resistance) compared to discussion section that is partially confusing and misleading.

Among the other things, the more evident in Material and methods section is that you did not mention gel electrophoresis analysis, then in Results section you insert the figure referring to it, that is by the way incomplete and the resolution is not high.

Have you evaluated a Manhattan plot to represent the genetic variants? Have you performed PCA (principal component analysis) to reduce the dimensionality of your dataset ? In general, usually, this type of genetic and bioinformatic analysis include them.

Why did you chose Study protocol as type of manuscript? I'd suggest a Communication form, in the event.

So, please, consider the reported observations and critical suggestions.

Please strongly revise English, reformulate sentences improving scientific data presentation. If possible, increment the number of animals (at least 100 animals) to make higher the statistical significance and the overall results.

Please check the format of Animals Journal in reporting references list.

Best regards.

Reviewer 4 Report

Anterior vena cava - it is a human terminology - probabby, better jugular vein.

Raw 143 - bostein et al[9]

Author Response

The anterior vena cava has been modified to jugular vein

Round 2

Reviewer 2 Report

Dear authors, you have improved the manuscript according to instructions, thus i recommend publishing paper in present version. 

Author Response

Dear Reviewer,

Thank you for considering publishing my article in [Animals]. I appreciate your valuable suggestions.

Kind regards,

Wu weiwei

Reviewer 3 Report

Dear Authors,

following the recommendations, suggestions and critical comments, the manuscript is really improved in the form, in the content and in the analyses. The certification of english revision is an added value. There are still some formatting and texting mistakes, so please check them. I suggest to remove the Table 1, beacuse superfluous and redundant, reporting information in the main text.

In my opinion, after a minor revision, the manuscript is prone and suitable for the publication as study protocol/article.

Best regards.

Author Response

Dear Reviewer,

Thank you for considering publishing my article in [Animals]. Thank you very much for your valuable suggestions. I made the following changes to the article.

  1. Corrected the wrong format and text in the article.
  2. Table 1 in the article is deleted.

Kind regards,

Wu weiwei

Reviewer 4 Report

The study provides genetic information about the Altay white-headed cattle 21 population, which can be used for future conservation and breeding research. 

Author Response

Dear Reviewer,

I appreciate your valuable suggestions.

Kind regards,

Wu weiwei